# Polyfluorinated crosslinker-based solid polymer electrolytes for long-cycling 4.5 V lithium metal batteries

Lingfei Tang[1,2,7], Bowen Chen[1,2,7], Zhonghan Zhang[3,4,7], Changqi Ma [1,2,7], Junchao Chen [5], Yage Huang[5], Fengrui Zhang[2], Qingyu Dong[2], Guoyong Xue[5], Daiqian Chen[2], Chenji Hu[5], Shuzhou Li [4], Zheng Liu [3,4], Yanbin Shen [1,2], Qi Chen [1,2] ✉ & Liwei Chen [2,5,6] ✉

Solid polymer electrolytes (SPEs), which are favorable to form intimate interfacial contacts with electrodes, are promising electrolyte of choice for long-cycling lithium metal batteries (LMBs). However, typical SPEs with easily oxidized oxygen-bearing polar groups exhibit narrow electrochemical stability window (ESW), making it impractical to increase specific capacity and energy density of SPE based LMBs with charging cut-off voltage of 4.5 V or higher. Here, we apply a polyfluorinated crosslinker to enhance oxidation resistance of SPEs. The crosslinked network facilitates transmission of the inductive electron-withdrawing effect of polyfluorinated segments. As a result, polyfluorinated crosslinked SPE exhibits a wide ESW, and the Li|SPE|LiNi$_{0.5}$Co$_{0.2}$Mn$_{0.3}$O$_2$ cell with a cutoff voltage of 4.5 V delivers a high discharge specific capacity of ~164.19 mAh g$^{-1}$ at 0.5 C and capacity retention of ~90% after 200 cycles. This work opens a direction in developing SPEs for long-cycling high-voltage LMBs by using polyfluorinated crosslinking strategy.

Lithium metal batteries (LMBs) have drawn significant attention due to high theoretical capacity (~3860 mAh g$^{-1}$) and low redox potential (−3.04 V vs. NHE) of lithium metal anode[1,2]. However, the growth of lithium dendrites will cause short-circuiting, which brings serious safety hazards with flammable liquid electrolytes[3–5]. Replacing a liquid electrolyte with a nonflammable solid electrolyte with high mechanical strength is effective in resisting the growth of lithium dendrites to improve the safety of LMBs[6–8].

Solid polymer electrolytes (SPEs), which exhibit superior plasticity and flexibility, have become promising solid electrolytes because they favor the formation of intimate interfacial contacts with

electrodes, which is a prerequisite to realize the long-cycling stability of solid-state LMBs[9,10]. The classic poly(ethylene oxide) (PEO)-based SPEs transport lithium ions through chain movement in the amorphous region, but their room-temperature ionic conductivity is not ideal[11,12]. Plastic crystals and even liquid electrolytes have been added to improve the conductivity, but these additives also bring undesired features such as low mechanical strength and inferior thermal stability[13–15]. The advantages and disadvantages of these additives are similarly present in ionic liquid (IL) based SPEs[16,17]. Recently, the conductivity of SPEs showed a significant improvement by tuning the polymer network structure to optimize the Li$^+$ pathway,

[1]School of Nano-Tech and Nano-Bionics, University of Science and Technology of China, Hefei 230026, China. [2]i-Lab, Suzhou Institute of Nano-Tech and Nano-Bionics, Chinese Academy of Sciences, Suzhou 215123, China. [3]CINTRA CNRS/NTU/THALES, UMI 3288, Research Techno Plaza, Singapore 637553, Singapore. [4]School of Materials Science and Engineering, Nanyang Technological University, 50 Nanyang Ave, Singapore 639798, Singapore. [5]School of Chemistry and Chemical Engineering, In situ Center for Physical Sciences, Shanghai Electrochemical Energy Device Research Center, and Frontiers Science Center for Transformative Molecules, Shanghai Jiao Tong University, Shanghai 200240, China. [6]Solid-State Battery Research Center, Global Institute of Future Technology, Shanghai Jiao Tong University, Shanghai 200240, China. [7]These authors contributed equally: Lingfei Tang, Bowen Chen, Zhonghan Zhang, Changqi Ma. ✉e-mail: qchen2011@sinano.ac.cn; lwchen2018@sjtu.edu.cn

which achieved over 1 mS cm$^{-1}$ at 25 °C even without any semi-solid or liquid additives[18–20].

A wide electrochemical stability window (ESW) is a prerequisite for any solid-state electrolytes to incorporate high-voltage cathode materials in battery design to deliver high specific capacity and energy density. However, SPEs containing these easily oxidized oxygen-bearing groups usually exhibit a narrow ESW[21]. As a result, the SPEs suffered from serious oxidation at a high potential, leading to inferior cycle stability. As an example of solving this problem in polycarbonate electrolytes, the easily oxidized cyclic carbonate was ring-opened and further polymerized by organometallic catalysts, achieving an ESW ~4.8 V and conductivity ~1.1 mS cm$^{-1}$ at 25 °C[22]. A different approach using fluorine-containing groups, which has a strong electron-withdrawing effect, has also been introduced. Introducing fluorine-containing groups in monomers have enhanced the oxidation resistance of polar groups and led to a wide ESW of over 5 V[23–25]. Nevertheless, a wide ESW cannot be directly translated into a high cut-off voltage of long-cycling LMBs. ESW is consistently overestimated with respect to the electrochemical stability in practical cells. On the one hand, the ESW is usually measured by linear sweep voltammetry (LSV) with flat blocking electrodes, which shows lower reactivity compared with composite cathodes with high surface area and carbon conductive additive. On the other hand, cathode materials with transition metals may exhibit specific catalytic activity, which further exacerbates oxidation. Up to now, SPEs that adapt to long-cycling LMBs with a cutoff voltage of 4.5 V or higher are yet to be demonstrated.

In this manuscript, we apply a polyfluorinated crosslinking agent to enhance the oxidation resistance of SPEs. An SPE is prepared via ultraviolet (UV) light-initiated copolymerization of pyrrole-based IL, vinyl ethylene carbonate (VEC) monomers, and a polyfluorinated crosslinker. The resulted polyfluorinated crosslinked SPE exhibits a superior conductivity of 1.37 mS cm$^{-1}$ at 25 °C, a wide ESW of 5.08 V, and high mechanical strength. These properties allow the preparation of LMBs incorporating a LiNi$_{0.5}$Co$_{0.2}$Mn$_{0.3}$O$_2$ (NCM523) cathode at a charging cutoff voltage of 4.5 V. The assembled Li|SPE|NCM523 cell delivers a high discharge specific capacity of ~164.19 mAh g$^{-1}$ at 0.5 C and a superior capacity retention of ~90% after 200 cycles. The electron-withdrawing effect of polyfluorinated groups in the polymer network contributes to the improved electrochemical oxidation resistance. In addition, the crosslinked structure also strengthens the mechanical modulus of the SPE to resist the growth of lithium dendrites, which further enhances the long-cycling stability of the LMB.

## Results and discussion

### Preparation of solid polymer electrolytes

The SPE film ~100 μm in thickness consisted of three monomers of 1-allyl-1-methyl-pyrrolidinium bis(trifluoromethanesulfonyl) imide IL, VEC, and polyfluorinated crosslinker 2,2,3,3,4,4,5,5-octafluoro-1,6-hexanediol diacrylate (OFHDODA) blended with lithium salt (LiTFSI), and photoinitiator (IRGACURE 819) was prepared by solution casting and UV light curing sequentially, as schematically illustrated in Fig. 1a (see the Experimental Section for details). Supplementary Fig. 1 shows the colorless, transparent electrolyte turned from a liquid into a solid after UV light curing. The SPE with flaming-resistant IL is non-flammable, as shown in Supplementary Movie 1. Figure 1b shows the

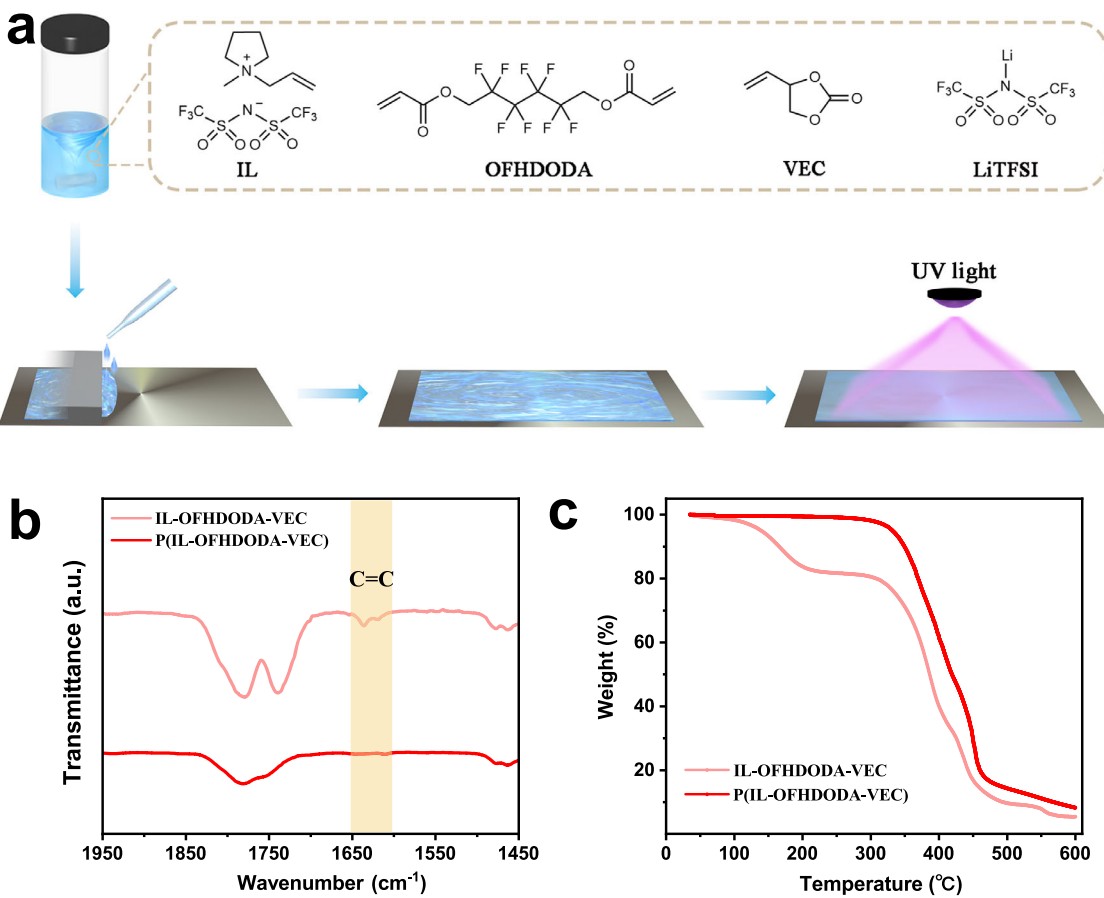

**Fig. 1 | Preparation of SPEs. a** Schematic illustration of the preparation of P(IL-OFHDODA-VEC). **b, c** FTIR spectra (**b**) and thermogravimetric curves (**c**) of IL-OFHDODA-VEC and P(IL-OFHDODA-VEC).

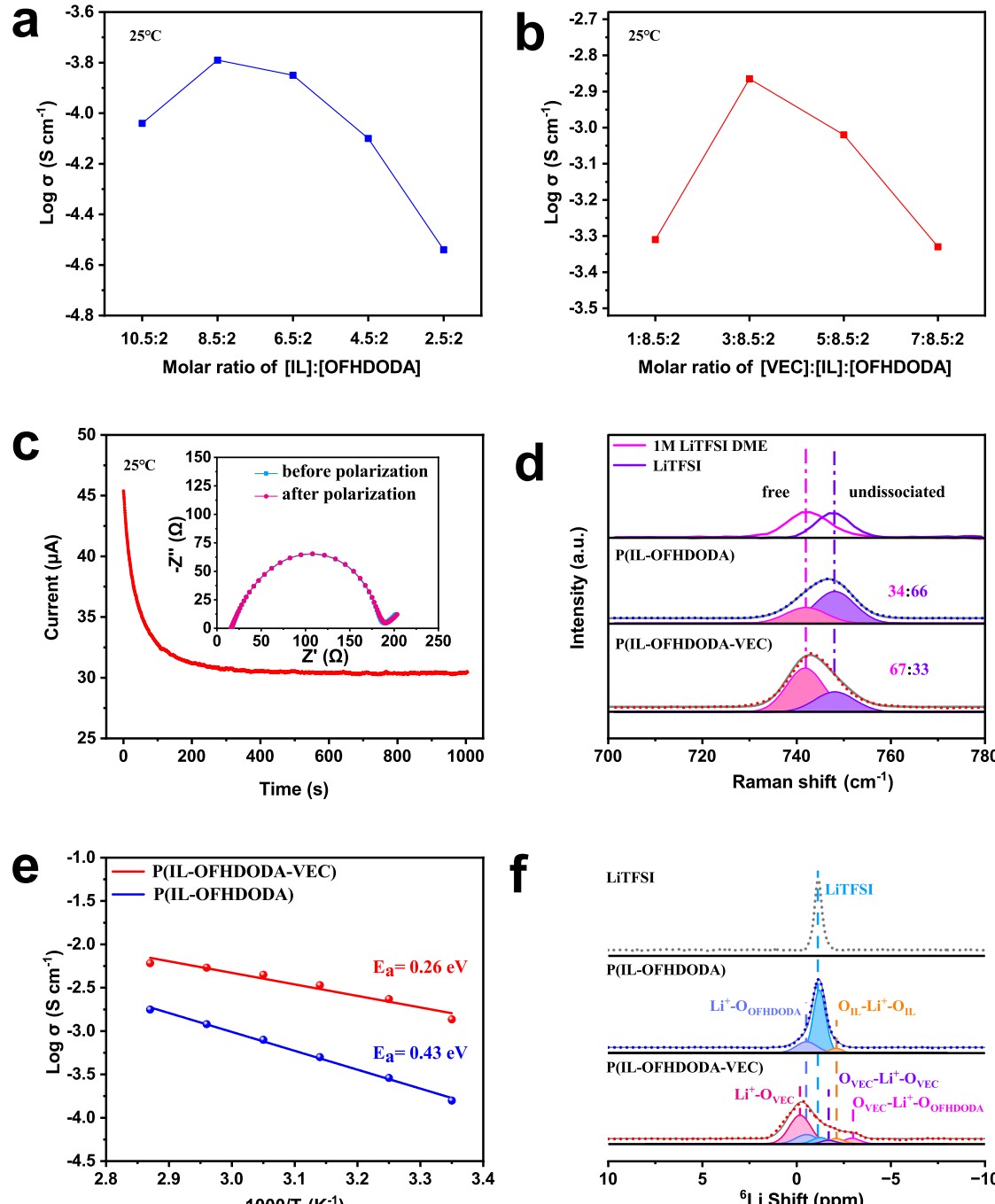

**Fig. 2 | Ionic transport properties of SPEs. a, b** Ionic conductivity of P(IL-OFH-DODA) (**a**) and P(IL-OFHDODA-VEC) (**b**) at 25 °C with different monomer molar ratios. **c** CA curve during polarization of Li|P(IL-OFHDODA-VEC)|Li symmetrical cell at 25 °C with total applied potential difference of 10 mV. Inset: EIS of Li|P(IL-OFHDODA-VEC)|Li symmetrical cell. **d** Raman spectra of P(IL-OFHDODA) and P(IL-OFHDODA-VEC). **e** Ionic conductivity of P(IL-OFHDODA) and P(IL-OFHDODA-VEC) as function of temperature range from 25 to 75 °C. **f** $^{6}$Li NMR spectrum of P(IL-OFHDODA) and P(IL-OFHDODA-VEC).

Fourier transform infrared (FTIR) spectra of the electrolyte before and after UV light curing. The peak at 1640 cm$^{-1}$ assigned to the stretching vibration of C=C bonds became invisible after UV light curing. Thermogravimetric analysis (TGA) of the electrolyte before and after UV light curing is shown in Fig. 1c. Before UV light curing, the electrolyte exhibited a significant mass loss of 16.6% between 125 and 200 °C. In contrast, the electrolyte presented a single decomposition temperature of 325 °C after UV light curing. Combined with both FTIR spectra and TGA, it demonstrated that IL, OFHDODA, and VEC monomers of the SPE were copolymerized by the C=C bonds after UV

light curing, which was confirmed by a minor weight loss of ~3.1 % due to oligomers extracted by diethyl ether (Supplementary Fig. 2).

### Ionic conductivity of solid polymer electrolytes

The ionic conductivity of the SPE was sensitive to the ratio of each monomer. Here, a molar ratio of [LiTFSI] to [the monomers] was set as 1:2. The molar ratio of [IL] to [OFHDODA] was tuned from 10.5:2 to 2.5:2 at 25 °C, which reached a maximum ionic conductivity of 0.16 mS cm$^{-1}$ under 8.5:2 (Fig. 2a). The ionic conductivity further increased from 0.49 to 1.37 mS cm$^{-1}$ by incorporating VEC with a molar ratio of

[IL]:[OFHDODA] to [VEC] from 8.5:2:1 to 8.5:2:3 at 25 °C (Fig. 2b). However, the ionic conductivity decreased with a higher VEC molar ratio. The conductivity of P(IL-OFHDODA-VEC) with an optimized molar ratio is comparable to that of state-of-the-art SPEs[18–20,22,26]. The Li$^+$ transference number ($t_{Li}^+$) ~0.40 at 25 °C is the ratio of Li$^+$ transport to the total ions (cations and anions) that transport in SPEs, which can be obtained from chronoamperometry (CA) and electrochemical impedance spectrum (EIS) tests (Fig. 2c). The superior conductivity and decent Li$^+$ transport number of SPEs will benefit a high capacity and superior rate performance of LMBs.

The effects of VEC on the superior ionic conductivity of the SPE have been investigated via Raman spectroscopy, which is sensitive to the bonding state of TFSI$^-$[27]. Figure 2d shows Raman spectra of stretching vibrations of N–S bonds in TFSI$^-$ with different SPE compositions. Specifically, the Raman shifts of N–S bonds in 1 M LiTFSI DME and LiTFSI crystals are regarded as references for dissociation and undissociated LiTFSI salt states, respectively, which guide the fitting of the spectra of P(IL-OFHDODA) and P(IL-OFHDODA-VEC) to distinguish different states of TFSI$^-$ (Fig. 2d). The ratio of free state TFSI$^-$ to undissociated state TFSI$^-$ in the SPE increases from 34:66 without VEC to 67:33 with VEC. This result indicates that more dissociation state LiTFSI of P(IL-OFHDODA-VEC) than that of P(IL-OFHDODA) can be attributed to the interaction between VEC and Li$^+$, leading to higher ionic conductivity.

The superior ionic conductivity of the SPE with VEC was also attributed to the low Li$^+$ transport energy barrier, as shown in Fig. 2e. The activation energy ($E_a$) of P(IL-OFHDODA) and P(IL-OFHDODA-VEC) extracted from the temperature dependence of ionic conductivities from 25 to 75 °C was 0.43 and 0.26 eV, respectively. The thermal transition was investigated by differential scanning calorimetry (DSC, Supplementary Figs. 3, 4). P(IL-OFHDODA-VEC) exhibited a lower glass transition temperature ($T_g$) ~ −63 °C than that of P(IL-OFHDODA) ~ −59 °C. The dielectric spectra of P(IL-OFHDODA-VEC) and P(IL-OFHDODA) was also analyzed as described in Supplementary Note 1. P(IL-OFHDODA-VEC) presented a shorter segment relaxation time ($\tau$) and lower apparent activation energy (B) than that of P(IL-OFHDODA) (Supplementary Table 1). Combined with both DSC and dielectric spectra, it demonstrated that VEC can plasticize the P(IL-OFHDODA) to activate segment movement and increase the free volume for a low Li$^+$ transport energy barrier. However, crystallization of VEC occurred in excessive amounts, which was evidenced by a broad diffraction peak around 30° in XRD spectra (Supplementary Fig. 5), resulting in suppressed segment movement and reduced ionic conductivity as consistent with a higher $T_g$[20].

The effects of VEC on the low Li$^+$ transport energy barrier of the SPE have been further explored via $^6$Li solid-state nuclear magnetic resonance (NMR), which is sensitive to the local chemical environment of Li$^+$[28,29]. The $^6$Li NMR spectra of P(IL-OFHDODA) and P(IL-OFHDODA-VEC) are shown in Fig. 2f. For P(IL-OFHDODA), three peaks representing different lithium environments were identified by fitting the $^6$Li NMR spectrum. Among them, the light blue peak at −1.2 ppm has the largest proportion of ~73%, which has the same chemical shift as the $^6$Li NMR peak of the LiTFSI crystal. For P(IL-OFHDODA-VEC), five peaks representing different chemical environments of lithium were identified by fitting the $^6$Li NMR spectrum peaks. Compared to the $^6$Li NMR spectrum of P(IL-OFHDODA), three peaks emerge at −0.2, −1.7, and −2.9 ppm. Among them, the red peak at −0.2 ppm has the largest proportion of ~55%, which indicates that a larger proportion of Li$^+$ changes from strongly bound states to weakened bound strength.

Since the interaction between Li$^+$ and electron-withdrawing atoms of the functional groups play an important role in the chemical environment of Li$^+$, density functional theory (DFT) calculations of the adsorption energy of Li$^+$ on different monomers or dimers were carried out. The results are listed in Supplementary Table 2 and optimized geometry models are presented in Supplementary Fig. 6. In general, the Li$^+$ adsorbed around the O atom in the VEC side chain (Li$^+$-O$_{VEC}$) exhibited the weakest binding affinity.

As the weak binding of Li$^+$ on the O atom in the SPE segment generates a small chemical shift of Li away from 0 ppm, the red fitting peak of $^6$Li NMR of P(IL-OFHDODA-VEC) possessed the largest proportion is believed to be the result of the interaction between Li$^+$ and the O atom in the VEC side chain (Li$^+$-O$_{VEC}$)[30]. This suggests that the incorporation of VEC into the SPE would contribute to the dissociation of LiTFSI, which is consistent with the results obtained by the Raman spectroscopy. As a result, the interaction of Li$^+$ with the VEC side chain inside the SPE is beneficial to achieve high ionic conductivity at 25 °C.

Overall, to realize a superior ionic conductivity, the SPE will not only effectively dissolve LiTFSI and then dissociate Li$^+$ from TFSI$^-$, but also efficiently coordinate Li$^+$ and promote its mobility. The high polarity IL offers good solubility of LiTFSI[31]. The interaction between the O atom in the VEC and Li$^+$ effectively dissociates Li$^+$ from TFSI$^-$ as proved by the Raman spectra. The ester groups show high affinity with Li$^+$, as supported by DFT calculations and solid-state NMR spectra. Thus the ester groups are believed to be the hopping site for Li$^+$, as the Li$^+$ coupling/decoupling process is likely to take place around them. The segment movement capability can be maintained to prevent overcrosslinking and VEC crystallization, which is evidenced by XRD, DSC, and dielectric spectra. Therefore, the ratio of IL, VEC, and OFHDODA monomers needs to be deliberately adjusted to achieve an optimized condition.

## Electrochemical stability of solid polymer electrolytes

The ESW of the SPE, which is a critical factor in working with a high-voltage cathode, was investigated. Figure 3a shows the LSV of an asymmetrical cell of Li|P(IL-OFHDODA-VEC)|carbon at 25 °C, wherein ESW ~5.08 V (vs. Li$^+$/Li) is read as the voltage at which the oxidation current exceeds 1.5 μA. Here, the porous carbon electrode rather than the flat blocking electrode was used to avoid overestimating the ESW, which has a high contact area that is close to composite cathodes. To understand the origin of the superior ESW, LSVs of PIL, POFHDODA, and PVEC and each pair of them were measured as shown in Supplementary Fig. 7. Figure 3b shows that the extracted ESW of PIL, POFHDODA, PVEC, P(IL-VEC), P(IL-OFHDODA) and P(OFHDODA-VEC) is 5.51, 5.13, 4.35, 4.39, 5.31, and 5.12 V (vs. Li$^+$/Li), respectively. Compared with PIL and POFHDODA, PVEC possessed the narrowest ESW. Nevertheless, the ESW of P(OFHDODA-VEC), i.e. VEC copolymerized with OFHDODA, was close to that of P(IL-OFHDODA-VEC). Moreover, ESW of P(IL-OFHDODA-VEC) increased with a higher concentration of OFHDODA, as seen in Supplementary Table 3. All these results demonstrate that OFHDODA plays an important role in improving the electrochemical stability of SPE.

The effect of OFHDODA on the ESW of the SPE can be understood from the molecular structure. OFHDODA possesses both the difluoro methylene chain and the ability to crosslink. From DFT calculation, the difluoro methylene chain in OFHDODA has a strong electron-withdrawing effect, which effectively reduces the electron density of its neighbor ester groups to enhance the overall oxidation resistance, especially after copolymerized with VEC groups (Supplementary Figs. 8, 9). After being coupled with VEC groups, the electron-withdrawing effect from the polyfluorinated group can suppress the frontier orbital electron from concentrating on the ester group but also distributes on VEC segments. Delocalization of frontier electrons is helpful for avoiding easy oxidation on a single reaction position. Such behavior is absent in the nonfluorinated crosslinker HDODA site. The HOMO is concentrated solely on ester groups after VEC copolymerization for HDODA. As a second proof, the required energy to remove an electron from the system for VEC copolymerized OFHDODA ~9.19 eV is higher than that for VEC copolymerized HDODA ~8.97 eV (Supplementary Table 4). Furthermore, the ESW of SPE with HDODA

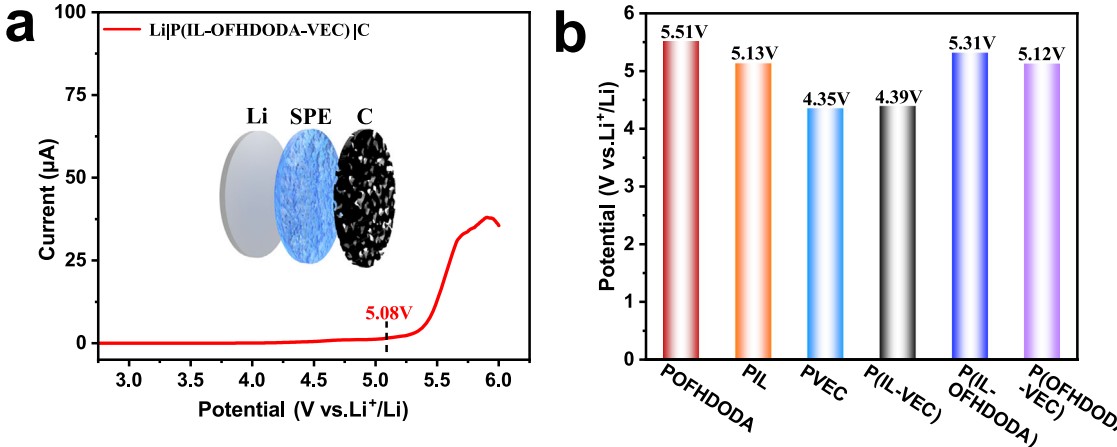

**Fig. 3 | ESW of SPEs. a** LSV curve of Li|P(IL-OFHDODA-VEC)| carbon. **b** ESW of PIL, POFHDODA, PVEC, P(IL-OFHDODA), P(IL-VEC), and P(OFHDODA-VEC).

reduced to 4.55 V (vs. Li[+]/Li, Supplementary Fig. 10) is another proof of the stabilization effect of the polyfluorinated OFHDODA group.

The effect of the ability of crosslinking is also studied. After replacing OFHDODA with fluorinated non-crosslinkable OFPMA, it is observed that the frontier electron is concentrated on one of the VEC segments after copolymerization, as seen in Supplementary Fig. 9, suggesting less stability than the OFHDODA case. The energy requirement of removing an electron from the system agrees with a value of 8.91 eV for VEC copolymerized OFPMA (Supplementary Table 4). The ESW of SPE with OFPMA ~4.71 V (vs. Li[+]/Li) was slightly higher than that with HDODA but lower than that with OFHDODA (Supplementary Fig. 10). The non-crosslinking nature might work in opposition to stabilization, as all the copolymerization has to take place on one side of the molecule, resulting VEC segments easier be attacked than the ester group of OFPMA that being surrounded by VEC segments. In addition, the crosslinked structure is more bulky and less linear-shaped. This may inhibit sidechain movement and reduce the possibility of contact between oxygen-containing polar groups and electrodes-induced oxidation, as supported by a higher $T_g$ in DSC, higher B, and longer τ in dielectric spectra (Supplementary Fig. 4 and Table 5).

To further investigate the universality of polyfluorinated crosslinking on improving ESW of SPEs, some other SPEs containing oxygen-bearing polar groups with and without OFHDODA have been tested. As shown in Supplementary Fig. 11, these SPEs all exhibited wider ESW after polyfluorinated crosslinking.

**Anode interface compatibility**

The stability of the SPE to lithium metal plays an important role in determining the cycle life of LMBs. Figure 4a shows periodically charged and discharged curves at a charge density of 0.1 mAh cm[−2] in Li||Li symmetric cells. The SPE with OFHDODA crosslinking agent-based cell exhibited a flat polarization curve with an average constant polarization as low as 43 mV for up to 2500 h. In contrast, the SPE without OFHDODA-based cell short-circuited after 388 h. Supplementary Fig. 12 shows the voltage response of P(IL-OFHDODA-VEC) from 0.1 mAh cm[−2] to 0.8 mAh cm[−2], which maintains excellent interface compatibility even under a charge density as high as 0.8 mAh cm[−2].

To understand the improvement of interface stability between the Li and the SPE by adding OFHDODA, the SEM images of lithium deposition morphology on the Li from disassembled Li|P(IL-OFHDODA-VEC)|Li and Li|P(IL-VEC)|Li cells after cycling were compared with each other (Fig. 4b). It is obvious that lithium deposited unevenly and dendrites are formed when OFHDODA is not added. For Li|P(IL-OFHDODA-VEC)|Li, however, the morphology of the lithium deposition was rather uniform without lithium dendrites. Moreover,

nanomechanical images of P(IL-VEC) and P(IL-OFHDODA-VEC) were measured by atomic force microscopy (AFM). As seen in Fig. 4c, Young's modulus is uniformly distributed, and P(IL-OFHDODA-VEC) ~142 MPa is much higher than that of P(IL-VEC) ~27 MPa. Since the information obtained by AFM came from the sample surface, the bulk mechanical properties were further measured. Compared with P(IL-VEC), P(IL-OFHDODA-VEC) exhibited not only a higher bulk Young's modulus, but also a higher bulk shear modulus as shown in Supplementary Table 6 and Supplementary Fig. 13. These results demonstrated that the mechanical strength of the SPE improved significantly by the introduction of the crosslinking agent, which was important to maintain the interface over the course of lithium plating and stripping[32,33].

The cycling stability of polymer electrolytes against Li also depends on the solid electrolyte interphase (SEI) properties of the Li| SPE interface. X-ray photoelectron spectroscopy (XPS) was conducted to investigate the surface compositions of the Li metal before and after long-term cycling. As shown in Fig. 4d, the F 1s spectrum of the Li metal exhibits no obvious fluorine product before cycling, but a significant Li-F signal is found at 685.5 eV after cycling[34]. Meanwhile, the S 2p spectrum of the Li metal shows no obvious signal, while six peaks at 160.5, 161.7, 166.0, 167.9, 168.6, and 169.3 eV ascribed to the S $2p_{3/2}$ characteristic peaks of $Li_2S$, $Li_2S_2$, $RSO_2Li$, $RSO_3Li$, $Li_2SO_3$, and $Li_2SO_4$, respectively, emerge after cycling (Fig. 4e)[35,36]. The N 1s spectrum shows that two peaks at 398.4 and 401.1 eV reflect the composition of $Li_2N-SO_2^-$ and $Li_3N$ after cycling[37]. In contrast, the Li metal exhibits no obvious $Li_3N$ before cycling (Fig. 4f).

The evidence that the species containing sulfur, e.g., $Li_2S$, etc., appear after cycling suggests that the stable SEI layer mainly originates from the decomposition of TFSI[−]. Notably, the characteristic peaks of $Li_3N$ and LiF indicate that they also participate in the construction of a stable SEI. The presence of $Li_3N$ and $Li_2S$ in the SEI layer has also been reported to have beneficial effects for fast Li[+] ion transfer[38,39]. LiF in the SEI layer is effective in suppressing the continuous decomposition of the SPE induced by the interfacial electrochemical reaction[40]. Therefore, the SEI containing $Li_3N$, $Li_2S$, and LiF is beneficial to inhibit the further decomposition of the SPE and ensure uniform current density during cycling.

**Cathode interface compatibility**

The interface compatibility between SPE and cathode also plays an important role in determining the cycling stability of LMBs, especially at a high cutoff voltage. Here, NCM523 was selected to assemble the cell due to its superior stability at a high cutoff voltage of ~4.5 V. The morphology of the NCM523 cathode particles was observed by high-resolution transmission electron microscopy (HRTEM), and it is found

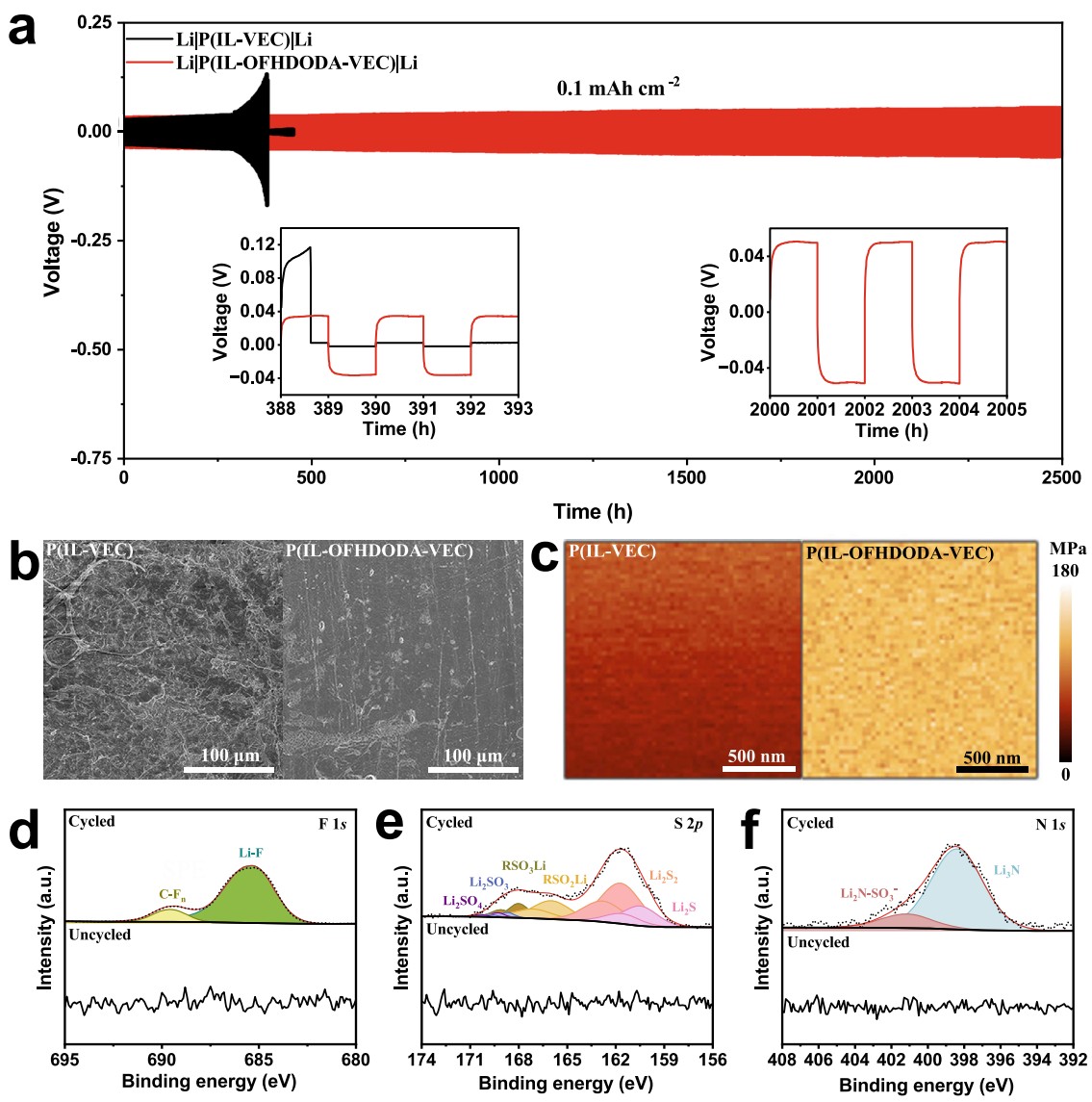

**Fig. 4 | Characterizations of anode interface evolution. a** Galvanostatic cycling of lithium symmetrical cells with P(IL-VEC) and P(IL-OFHDODA-VEC) at 0.1 mAh cm⁻². **b** SEM images of Li metal with P(IL-VEC) and P(IL-OFHDODA-VEC) after cycling. **c** Young's modulus of P(IL-VEC) and P(IL-OFHDODA-VEC). **d–f** XPS analysis of F 1s (**d**), S 2p (**e**), and N 1s (**f**) spectrum of the Li metal before and after 2500 h of cycling.

that there is an amorphous cathode electrolyte interphase (CEI) layer of 2 nm before cycling, which increases to 4 nm after 200 cycles at a cutoff voltage of 4.5 V (Fig. 5a). Combined with time-of-flight secondary-ion mass spectrometry (TOF-SIMS) (Fig. 5b) and energy dispersive X-ray spectroscopy (EDS) (Supplementary Fig. 14), it is confirmed that the CEI layer contained a large amount of LiF and gradually became homogeneous and dense with cycling, resulting in a slight increase in interfacial impedance (Supplementary Fig. 15). The LiF-rich CEI layer not only has good electronic insulation but also avoids the direct contact of the SPE with the cathode surface, which can prevent the high valence nickel from catalyzing the further decomposition of electrolyte under high voltage[41,42].

To understand the formation mechanism of LiF, XPS was employed to investigate the change in the composition of the NCM523 cathode before and after cycling. The F 1s spectrum exhibits a higher peak intensity at 685.5 eV corresponding to Li-F after cycling than that before cycling (Fig. 5c), which is consistent with the increased thickness of LiF as measured by TOF-SIMS and EDS. In addition, the peak intensity of the C 1s spectrum at 293.3 eV corresponding to $CF_3$ decreases, and the N 1s spectrum at 401.1 eV corresponding to

$Li_2N-SO_2^-$ appears after cycling (Figs. 5d, e). In contrast, the C 1s spectrum also reveals that the characteristic peaks of C-C/C-H, C-O, $CH_2-CF_2$, C=O, and $CF_2$ at 284.8, 285.7, 287.6, 288.5, and 290.4 eV, respectively[43], show little change before and after cycling.

The formation of LiF before cycling may be due to the photo-induced substitution reaction between F-containing radicals and LiOH on the cathode surface. The increased thickness of LiF after cycling originates from the decomposition of TFSI⁻, which is consistent with the decreased C 1s peak at 293.3 eV corresponding to the $CF_3$ group of TFSI⁻[44]. Meanwhile, the characteristic peaks of C-C/C-H, C-O, $CH_2-CF_2$, C=O, and $CF_2$ exhibit little change before and after the cycle, indicating that the polymer main chain and side chain maintain high stability. In addition, the spectrum of N 1s shows that TFSI⁻ is decomposed to $Li_2N-SO_2^-$ after cycling. However, the C-N⁺ of the pyrrole side chain in the electrolyte does not change significantly, which further indicates that the polymer is stable (Fig. 5e).

**Performance of solid polymer electrolyte-based Li metal cells**
Based on the SPE with the high ionic conductivity, wide ESW, superior mechanical strength, and excellent interface compatibility,

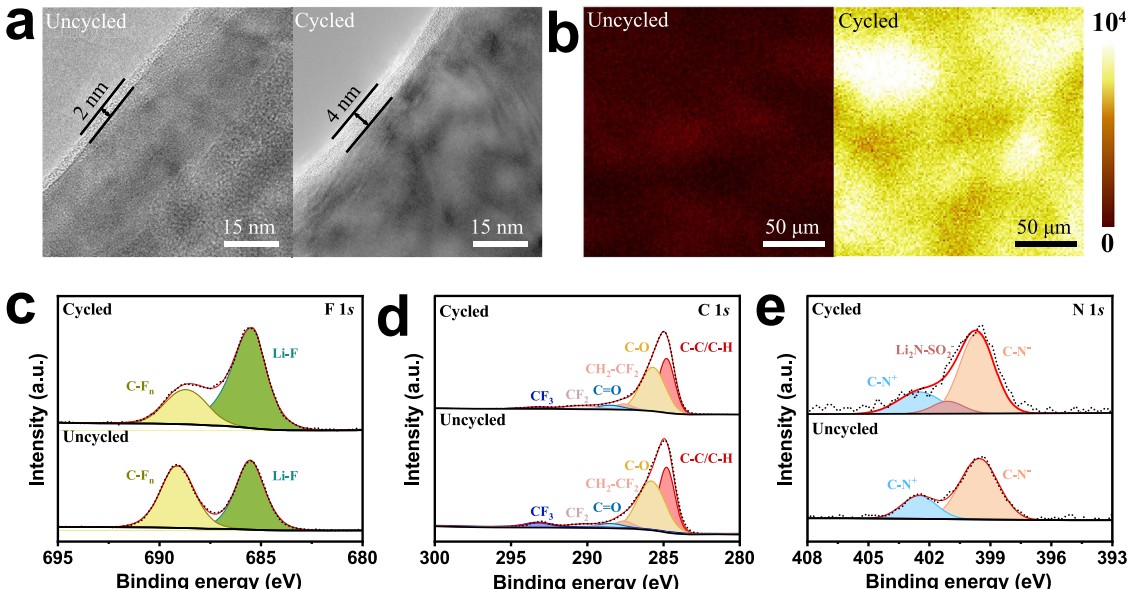

**Fig. 5 | Characterizations of cathode interface evolution. a** HRTEM images of NCM523 cathode particles before and after 200 cycles. **b** 2D TOF-SMIS measured LiF images of NCM523 cathode surface before and after 200 cycles. **c**–**e** XPS analysis of F 1s (**c**), C 1s (**d**), and N 1s (**e**) spectrum of the NCM523 cathode before and after 200 cycles.

Li|P(IL-OFHDODA-VEC)| NCM523 full cell (1 C = 180 mAh g⁻¹) were assembled to evaluate electrochemical performance. Figure 6a, b shows the galvanostatic charge–discharge curves of the full cell with a cutoff voltage of 4.5 V at 30 °C, which exhibits a high initial discharge capacity of 164.19 mAh g⁻¹ at 0.5 C and stable cycling performance with a discharge capacity of 146.96 mAh g⁻¹ even after 200 cycles. To the best of our knowledge, this is one of the fewest polymer electrolytes with self-supporting properties for 4.5 V class LMBs with a high capacity retention of ~90% after 200 cycles (Supplementary Table 7). Figure 6c shows the rate performance of the full cell as the current densities increased from 0.1 C to 2 C. The full cell exhibited a superior specific capacity of 185.80 mAh g⁻¹ at 0.1 C, and ~104.82 mAh g⁻¹ at 2 C. Therefore, the full cell based on the SPE with an optimized monomer ratio can achieve a superior initial discharge capacity, cycling stability, and rate retention simultaneously, as detailed in Supplementary Table 3.

The universality of the SPE was further confirmed by assembling Li|P(IL- OFHDODA-VEC)|LiFePO₄ (LFP) full cell (1 C = 165 mAh g⁻¹). The LFP full cell with a cutoff voltage of 4.0 V at 30 °C exhibits a high max discharge capacity of 161.20 mAh g⁻¹ at 0.5 C and retains a discharge capacity of 140.72 mAh g⁻¹ even after 600 cycles (Supplementary Fig. 16). The LFP full cell also exhibits a superior rate performance with a specific capacity of 166.18 mAh g⁻¹ at 0.1 C, and ~122.28 mAh g⁻¹ at 2 C (Supplementary Fig. 16).

Li|P(IL-OFHDODA-VEC)|NCM523 pouch cell have been assembled for practical application exploration (Fig. 6d–f). The pouch cell can simultaneously light up a SINANO logo composed of 70 blue light-emitting diode (LED) lamps connected in parallel. When the pouch cell was cut or bent in a fully charged state, it can still light up the logo, indicating high safety and superior mechanical stability.

In summary, a SPE has been prepared via UV light-initiated radical polymerization of pyrrole-based ionic liquid, vinyl ethylene carbonate monomers, and a polyfluorinated crosslinker. The resulted SPE possessed a high ionic conductivity of 1.37 mS cm⁻¹ at 25 °C, a wide ESW of 5.08 V, and a high mechanical strength. The polyfluorinated cross-linked SPE exhibited improved electrochemical oxidation resistance by not only the inductive electron-withdrawing effect of poly-fluorinated segments, but also the transmitted inductive effect through a crosslinked network. In addition, crosslinked structure

strengthened mechanical modulus to effectively resist the growth of lithium dendrites. As a result, the assembled Li|SPE|NCM523 full cell with a cutoff voltage of 4.5 V delivered a high discharge specific capacity of 164.19 mAh g⁻¹ at 0.5 C and stable cycling performance with a specific capacity of 146.96 mAh g⁻¹ after 200 cycles. It is demonstrated that incorporating polyfluorinated crosslinking agent is a widely applicable strategy to achieve mechanically and electrochemically robust SPEs.

## Methods

### Solid polymer electrolyte preparation

Bis(trifluoromethane) sulfonamide lithium (LiTFSI), vinyl ethylene carbonate (VEC, 99%), and poly(ethylene glycol) monomethacrylate (PEGMA, 98.0%) were purchased from Sigma-Aldrich. 1-allyl-1-methyl-pyrrolidinium bis(trifluoromethanesulfonyl) imide (IL, 99.0%) was purchased from Shang-hai Cheng Jie Chemical Co., Ltd. 2,2,3,3,4,4,5,5-Octafluoro-1,6-hexanediol diacrylate (OFHDODA, 93.0%), 1,6-Hex-anediol diacrylate (HDODA, 85.0%) and 2,2,3,3,4,4,5,5-Octafluoropentyl methacrylate (OFPMA, 98.0%) were purchased from Tokyo Chemical Industry (TCI). Phenylbis(2,4,6-trimethylbenzoyl)phosphine oxide (IRGACURE 819, 99%) was purchased from Badische Anilin-und-Soda-Fabrik (BASF). Vinylene carbonate (VC, 99.9%) was purchased from Dodo Chemical Co., Ltd. P(IL-OFHDODA-VEC) was prepared by UV light curing. Typically, the mixed IL, OFHDODA, VEC monomer, and LiTFSI were stirred for 40 min at 25 °C. After IRGACURE 819 (0.6% molar weight of the monomers) was added and stirred for 20 min, this mixture was blade-cast onto a substrate, followed by a photocuring process via 30 W of 365-nm UV light. All of these processes were carried out in an argon-filled glove box.

### Solid polymer electrolyte characterization

FTIR spectra of the SPE were collected using a Thermo Scientific Nicolet 6700 spectrometer. TGA (TG/DTA-6300) experiments were carried out in the temperature range from 35 to 600 °C under a nitrogen and oxygen atmosphere at a heating rate of 10 °C min⁻¹. The molecular weight distribution of the extracted SPE was measured by the positive-ion mode of matrix-assisted laser desorption/ionization time-of-flight mass spectrometry instrument (MALDI-TOF-MS, Bruker FLEX). Raman spectra were collected by a Renishaw inVia Qontor

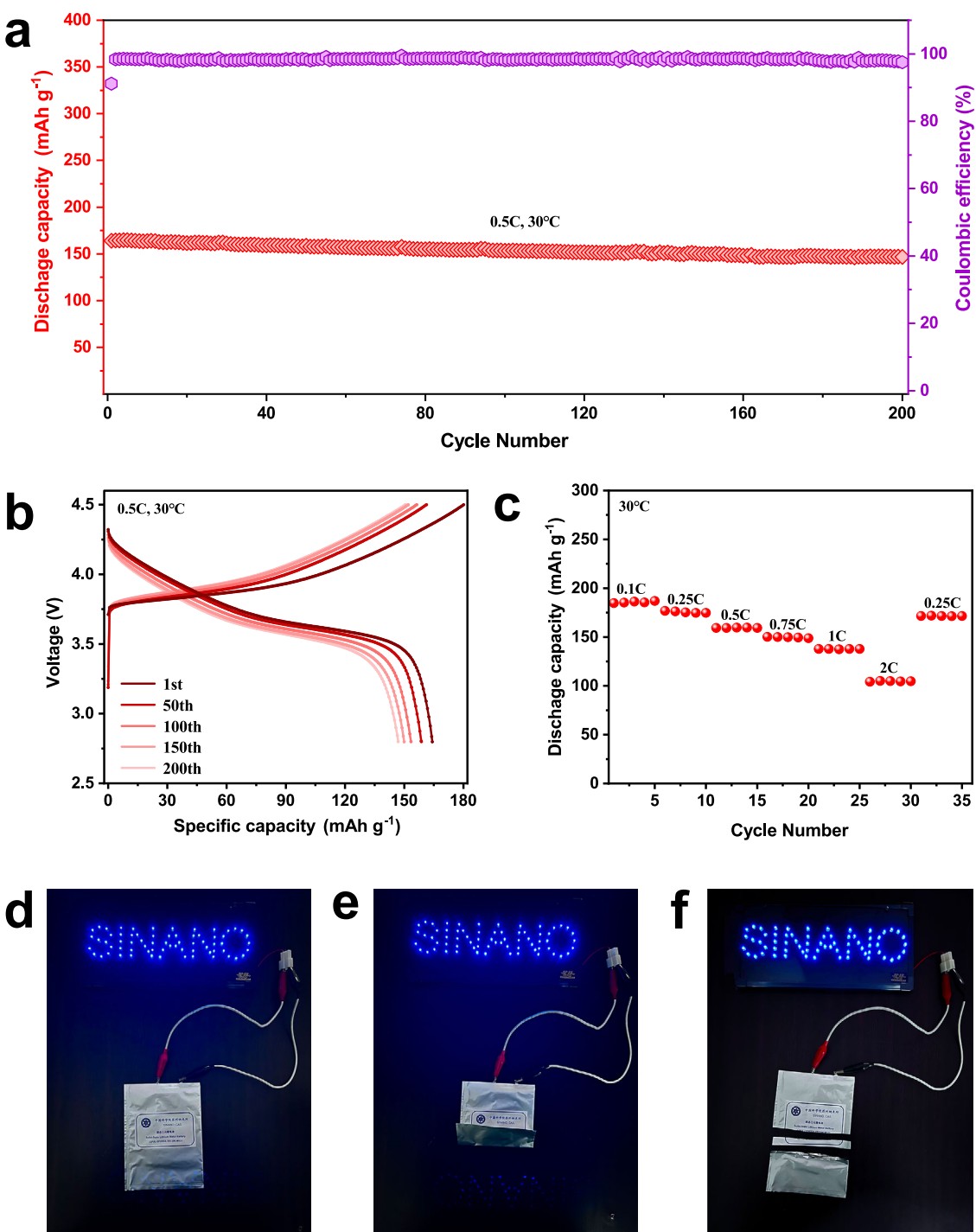

**Fig. 6 | Electrochemical performance of the full cell with the structure of Li|P(IL-OFHDODA-VEC)|NCM523.** **a** Cycling performance of Li|P(IL-OFHDODA-VEC)|NCM523 full cell at 0.5 C. **b** 1–200th charge and discharge curves of Li|P(IL-OFHDODA-VEC)|NCM523 full cell. **c** Rate performance of Li|P(IL-OFHDODA-VEC)|NCM523 full cell. **d**–**f** Photographs of fully charged Li|P(IL-OFHDODA- VEC)|NCM523 pouch cell to light up LED lamps before (**d**) and after folding (**e**) or cutting (**f**).

confocal Raman microscope using a 532-nm-wavelength laser. DSC analysis was executed in the range from −105 to 0 °C at a heating rate of 10 °C min⁻¹. XRD spectra were carried out on a Bruker D8 with Cu Kα radiation. $^6$Li static solid-state NMR spectra were recorded on a Bruker AVANCE III 400 equipped with an 89-mm wide-bore 9.4 T super-conducting magnet in 4.0 mm rotors at Larmor frequencies of 58.8 MHz. A recycle delay of 20 s was used for the accumulation of 1440 scans for every $^6$Li single pulse spectrum. The $^6$Li chemical shift was referenced to 0.1 M LiCl solution at 0 ppm. SEM measurements were performed by an FEI Quanta 400 FEG. Nanomechanical images

were obtained by a Cypher S AFM (Asylum Research, Oxford Instruments) with a Si probe (AC160TS, Olympus). Bulk mechanical properties were measured by KLA Nano Indenter G200 (Tencor Com.) and Instron 3365 universal testing machine. XPS measurements were performed by a Thermo Scientific ESCALAB Xi+ using a monochromatic Al Kα X-ray source (1486.6 eV) with a pass energy of 30 eV in a Vacuum Interconnected Nanotech Workstation (Nano-X). The HRTEM images were obtained using a TEM instrument (FEI Themis Z, 300 kV) equipped with EDX (Super X). TOF-SIMS measurements were performed via a TOF-SIMS5 (ION-TOF-GmbH) in Nano-X.

## Electrochemical measurements

A stainless steel symmetrical (SS|SPE|SS) cell was used to test the ionic conductivity and dielectric spectra. This test was performed via EIS experiments in a frequency range of 7 MHz to 100 mHz with an applied amplitude of 20 mV. The $t_{Li}^{+}$ was determined by the CA test on a Li symmetrical (Li|SPE|Li) cell with a potential of 10 mV until the current reached a steady state. The EIS measurements were taken before and after the polarization scans. The ESW was measured on an asymmetrical (Li|SPE|C) cell at a scan rate of 0.1 mV/s from OCV to 6 V. EIS, CA, and LSV tests were all carried out by an electrochemical working station (Biologic VMP-300).

The electrochemical performance of the SPE was evaluated based on CR2032-type coin cell, which were assembled with an NCM523 or LFP cathode, Li metal anode (99.9%, 400 μm, China Energy Lithium Co. LTD), and SPE (100–200 μm) in an Ar-filled glove box. For the cathode (diameter of 14 mm), NCM523 or LFP powders, acetylene black, and polyvinylidene fluoride (PVDF) mixed at a weight ratio of 8:1:1 in *N*-methyl-pyrrolidone (NMP) were coated on Al foil (>99.3%, 16 μm, Hefei Kejing Material Technology Co., LTD) using a blade with a mass loading of 0.4-1.1 mg cm$^{-2}$. Galvanostatic cycling tests were conducted on a Neware BTS battery test system (NEWARE technology Ltd. Shenzhen, China) using Li|P(IL-OFHDODA-VEC)|LFP/NCM523 coin cell within a potential range of 2.5–4.0 and 2.8-4.5 V (vs. Li$^{+}$/Li), respectively. The temperature of testing was 30 °C in a climate chamber.

## Density functional theory calculations

DFT calculations were carried out under the level of theory of the B3LYP functional[45,46] with D3BJ dispersion corrections[47] in conjunction with the 6-311 g(d,p) basis set[48]. The molecular models built for calculation (monomers or dimers without C=C bond) aimed to represent the side chain structures in polymer electrolytes. Computations of all monomers or dimers, and their Li$^{+}$ adsorbed configurations were performed with geometry optimization, followed by single point energy calculation for accurate adsorption energy and geometry comparison. The computations were performed with Gaussian 16a software[49], and molecular orbital information extraction and analysis were performed via Multiwfn[50].

## Data availability

The data that support the plots within this paper and other findings of this study are available from the corresponding author on reasonable request. The Source data generated in this study are provided in the Source Data file. Source data are provided with this paper.

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

## Acknowledgements

This work was supported by the Ministry of Science and Technology of China (Grant No. 2021YFA1202802 (Q.C.)), the National Natural Science Foundation of China (Grant Nos. 21875280, 21991150, 21991153, and 22022205 (Q.C. and L.C.)), the CAS Project for Young Scientists in Basic Research (YSBR-054 (Q.C.)), the Special Foundation for Carbon Peak Neutralization Technology Innovation Program of Jiangsu Province (No.BE2022026 (Q.C.)), and Ministry of Education of Singapore Tier 3 Program (Grant No. MOE-T2EP10220-0005 (S.L.)). The authors are grateful for technical support on XPS and TOF-SIMS measurements from Nano-X, SINANO. The DFT calculation was performed with NTU HPC resources.

## Author contributions

Q.C. conceived the idea. Q.C., B.C., and L.T. designed the experiments. Q.C. and L.C. supervised the project. Z.Z. performed DFT calculations supervised by S.L. and Z.L. J.C. measured NMR. Y.H. analyzed dielectric spectra. Q.D. and D.C. measured SEM. L.T. and B.C. prepared SPEs, assembled LMBs, and performed all other measurements. C.M., F.Z., G.X, C.H., and Y.S. assisted preparation of SPEs and assembly of cells. L.C., Q.C., B.C., and L.T. wrote the manuscript. All of the authors commented on the manuscript.

## Competing interests

The authors declare no competing interests.
