## [Peer Review File · Nature Communications]

Polyfluorinated Crosslinker-based Solid Polymer Electrolytes for Long-Cycling 4.5 V Lithium Metal BatteriesREVIEWER COMMENTS

Reviewer #1 (Remarks to the Author):

This paper reports high-performing solid polymer electrolytes enabled by polyfluorinated crosslinking for applications to high-potential batteries. Overall, the characterization and practical application of solid-state electrolytes are systematically investigated and the results look good. I think it should be published after minor revisions as listed below.

- 1) The Li transference number of the electrolytes is a little higher than that of a typical binary ionic conductor, which is likely due to the crosslinked nature of the polymer. I would suggest systematical investigation for some electrochemical properties, such as electrochemical window, battery cycling, and battery rate retention, of differing degree of crosslinking.
- 2) The authors insist that the addition of VEC facilitates the dissociation of Li⁺ cation from the LiTFSI. Based on their hypothesis, the ionic conductivity is supposed to keep increasing with the addition of VEC, but the high concentration of VEC leads to a decrease in ionic conductivity. Please rationalize it.
- 3) In Fig. 3a, it seems that the current value started to increase around 4.25 V, but the authors set the electrochemical stability window of 5.08 V. Please explain the criteria to set the stability window.
- 4) Please calculate the charge passed through from the Galvanostatic cycling in Fig. 4a, which would help to compare the result with the literature value.
- 5) The shear modulus is one of the important mechanical properties to maintain the interface over the course of plating and stripping. Please characterize the shear modulus of the electrolyte in addition to Young's modulus.
- 6) It seems the specific capacity of NCM523 batteries significantly decreased at 2C compared to that of LFP. More investigation is needed to prove the feasible application of the electrolyte to the high-potential cathode.

Reviewer #2 (Remarks to the Author):

This a well-written manuscript presenting analysis of novel solid polymer electrolyte formulation for high voltage Li-ion battery. Authors emphasize the use of polyfluorinated cross linkers that improve electrochemical stability and cycleability of electrolyte. The manuscript indeed presents extensive characterization of formulated material and its performance in the battery.

However, conceptually I do not see this approach any different to other approaches where fluorinated polymers were mixed in gel or solid polymer electrolytes. Many of the trends presented are expected, while some of the data, e.g. DFT calculations are just trivial and almost irrelevant. The binding energies of Li with segments in vacuum have little relevance to Li ion interaction in bulk environment. Of course configurations with two O atoms interacting with Li will be energetically more favorable than those with one. If authors wanted to get molecular scale understanding of Li transport mechanism they should consider atomistic molecular dynamics and conduct more systematic analysis of Li coordination, ion-ion correlations, etc. in representative condensed phase.

Manuscript provides plenty of data on cell performance, but not much basic characterization that would inform about mechanisms of observed improvement. For example, authors mentioned influence of fluorinated links on segmental mobility, but the only evidence provided is change in glass transition temperature by few degrees. Mechanical properties are only characterized by Young's modulus. Also DFT calculations confirming that presence of fluorinated segments improves oxidative stability and demonstrating the electron-withdrawing effect would be very useful.

I think this manuscript has interesting data and will be of interest to specialized community. But in my view it does not proposed conceptually new approach or mechanisms to justify publication as communication.

Reviewer #3 (Remarks to the Author):

In this study, authors developed a new solid polymer electrolyte composed of a fluorinated cross-linker, an ionic liquid-type monomer, and vinyl ethylene carbonate (VEC). As noted in the introduction, the high conductivity of PVECs (ex. *Nano Energy*, 2020, 73, 104786), and the improvement of oxidation resistance of SPEs by using fluorine monomers (ex. *Adv. Funct. Mater.* 2021, 31, 2101736), have already been reported. Therefore, in my opinion, the novelty of the paper is not enough high, as the concept of this study is kind of combination of previous studies. However, the high oxidation resistance as well as high ionic conductivity of this SPE achieved excellent battery performance comparable to those of state-of-the-art SPEs, as shown in Table S2. For this reason, the paper might be of interest to the readership of *Nature Communications*. However, major and minor points need to be addressed prior to publication.

1. Long-term cycling tests for symmetric Li cells were performed only at a very low current density of 0.1 mA/cm² despite the high ionic conductivity of the SPE. For practical application of the SPE toward Li metal batteries, cycling performance at higher current densities such as 1 mA/cm² should be investigated.

2. I don't understand why there is the optimal ratio for ionic conductivity among [IL], [VEC], and [OFHDODA]. In particular, the fluorine cross-linker would not contribute to ionic conductivity. Author should address the mechanism of the optimized monomer ratio. In addition, the role of the IL monomer should be clarified. The fluorine cross-linker contributes to the oxidative stability and the VEC to conductivity, then what is the role of the IL monomer?

3. It is surprising that the SPE shows such a high conductivity over 1 mS/cm. I am afraid that there are unreacted monomers that act as plasticizer of the SPE. Since the FT-IR-ATR could observe only a few micrometers of the surface of the sample, it might be an overstatement that all monomers are fully reacted. I think it is necessary to clarify the complete reaction of all monomers by other methods such as an extraction test of unreacted monomers using organic solvents.

4. For readers' information, the thickness of the SPE should be described.

5. Authors conducted the AFM measurement to investigate the elastic modulus of the SPE. However, the information obtained by AFM is the elastic modulus of the sample surface. Therefore, it would be better to investigate the bulk mechanical properties by tensile testing or DMA measurements.

REVIEWER COMMENTS

Reviewer #1 (Remarks to the Author):

This paper reports high-performing solid polymer electrolytes enabled by polyfluorinated crosslinking for applications to high-potential batteries. Overall, the characterization and practical application of solid-state electrolytes are systematically investigated and the results look good. I think it **should be published** after minor revisions as listed below.

1) The Li transference number of the electrolytes is a little higher than that of a typical binary ionic conductor, which is likely due to the crosslinked nature of the polymer. I would suggest systematical investigation for some electrochemical properties, such as electrochemical window, battery cycling, and battery rate retention, of differing degree of crosslinking.

Response: We appreciate reviewer's critical comment. Based on reviewer's suggestion, we systematically investigated the effects of crosslinking degree of SPE on electrochemical stability window (ESW), cycling stability, and rate retention.

Table A Summary of α , conductivity, ESW, full cell performance at different crosslinking degree

IL:OFHDODA:VEC	a	Conductivity mS cm ⁻¹	ESW V	t_{Li^+}	Initial discharge mAh g ⁻¹	Capacity retention %	Rate retention %		
							0.1C	2C	2C/0.1C
8.5:0:3	/	1.73	4.39	0.13	87.38	1%@40	187.23	2.93	2%
8.5:2:3	0.90	1.37	5.08	0.40	164.20	97%@40	184.14	101.32	55%
8.5:4:3	0.85	0.13	5.36	0.48	148.97	91%@40	174.95	43.66	25%
8.5:6:3	0.79	0.03	5.46	0.49	123.61	88%@40	161.52	35.52	22%

To reveal the change in crosslinking degree of the SPE with polyfluorinated crosslinker OFHDODA concentration, the dielectric spectra of segmental process at high frequencies has been analyzed based on the Havriliak – Negami (HN) function

after removing the contribution of ohmic conduction as expressed in Eq. 1 (J. Non-Cryst. Solids, 2002, 305, 40-49):

$$\varepsilon_{\text{HN}}^*(\omega) = \varepsilon_{\infty} + \frac{\Delta\varepsilon}{(1 + (i\omega\tau)^a)^b} \quad (1)$$

where $\varepsilon_{\text{HN}}^*(\omega)$ is the complex dielectric function, ε_0 and ε_{∞} are the low and high frequency dielectric constant, $\Delta\varepsilon$ is the dielectric intensity, τ is the relaxation time of segment movement, a and b are the fractional shape parameters that describe the symmetric and asymmetric broadening of the complex dielectric function when $a > 0$, $ab \leq 1$ holds. The a is inversely correlated with the crosslinking degree (Macromolecules, 1994, 27,1338-1343). Since a decrease from 0.90 to 0.79 with increased OFHDODA concentration as listed in Table A, it indicates that the increased crosslinking degree with a higher OFHDODA concentration.

The crosslinking degree dependent ESW, ionic conductivity, and t_{Li^+} of SPEs are all listed in Table A. The ESW of SPEs increases from 4.39 V to 5.46 V with increased crosslinking degree. The ionic conductivity of SPEs decreased from 1.73 mS cm⁻¹ to 0.03 mS cm⁻¹ with increased crosslinking degree. The Li⁺ transference number (t_{Li^+}) of SPEs increases from 0.13 to 0.49 with increased crosslinking degree.

The increased crosslinking degree with a higher OFHDODA concentration may inhibit sidechain movement, which reduces not only the possibility of contact between oxygen containing polar groups and electrodes induced oxidation, but also the capability of TFSI⁻ movement, leading to increased ESW and t_{Li^+} . However, Li⁺ movement is also suppressed to reduce the ionic conductivity.

The crosslinking degree dependent cycling and rate performance of full cells with a cutoff voltage of 4.5 V at 30 °C are all listed in Table A. With increased crosslinking degree, the initial discharge capacity at 0.5 C and capacity retention after 40 cycles were 87.38 mAh g⁻¹ and 1 %, 164.20 mAh g⁻¹ and 96 %, 148.97 mAh g⁻¹ and 90 %, 123.60 mAh g⁻¹ and 88 %, respectively. The corresponding specific capacity were 187.23, 184.14, 174.95, 161.52 mAh g⁻¹ at 0.1 C, and 2.93, 101.32, 43.66, 35.52 mAh g⁻¹ at 2 C, respectively.

The low ionic conductivity and/or t_{Li^+} of the SPE will exacerbate polarization loss in full cells, especially at high current densities, resulting in low initial discharge capacity and rate retention for non-crosslinked and over-crosslinked SPEs based full cells. While the SPE with an optimized crosslinking degree gives rise to both decent ionic conductivity and t_{Li^+} , benefiting superior initial discharge capacity and rate retention. At the same time, a superior cycling performance is achieved combined with a wide ESW and a high mechanical strength, which is attributed to improved interface stability.

In response to reviewer's critical comment, Table A has been added as Supplementary Table 3, the description of change in electrochemical window, battery cycling, and rate retention with SPE with different crosslinking degree has been added in Page 11 and 17 in the revised manuscript (in red font color) as follows:

Page 11: "Moreover, ESW of P(IL-OFHDODA-VEC) increased with a higher concentration of OFHDODA as seen in Supplementary Table 3. All these results demonstrate that OFHDODA plays an important role in improving the electrochemical stability of SPE."

Page 17: "Therefore, the full cell based on the SPE with an optimized monomer ratio can achieve a superior initial discharge capacity, cycling stability and rate retention simultaneously as detailed in Supplementary Table 3."

2) The authors insist that the addition of VEC facilitates the dissociation of Li^+ cation from the LiTFSI. Based on their hypothesis, the ionic conductivity is supposed to keep increasing with the addition of VEC, but the high concentration of VEC leads to a decrease in ionic conductivity. Please rationalize it.

Response: We thank reviewer's critical comment. In addition to the Li^+ dissociation capability, the ionic conductivity is also determined by the Li^+ movement ability, which is not monotonically increasing with VEC concentration.

The relaxation time (τ) of the segment movement can be obtained by a least square method fitting the HN-function (Eq.1) (J. Appl. Polym. Sci. 2022, 139, e52143). The τ decreased gradually when the molar ratio of [IL]:[OFHDODA] to [VEC] increased from 8.5:2:0 to 8.5:2:3 at -10 °C, indicating facilitated segment movement with a small amount of VEC. Then, the τ started to increase when the molar ratio of [IL]:[OFHDODA] to [VEC] further increased from 8.5:2:3 to 8.5:2:7, indicating suppressed segment movement with excessive VEC. Meanwhile, the apparent activation energy (B) of the segment movement can be extracted from the temperature dependent τ from -10 to -40 °C based on Vogel-Fulcher-Tammann (VFT) equation as expressed in Eq. 2:

$$\tau = \tau_0 e^{-B/(T-T_0)} \quad (2)$$

where τ_0 is the pre-exponential factor, T is the temperature, T_0 ($T_0 = T_g - 50K$) is the ideal glass transition temperature. The lowest B at the molar ratio of [IL]:[OFHDODA] to [VEC] of 8.5:2:3 indicates outstanding segment movement ability, which agrees with the corresponding smallest τ . The VEC concentration dependent segment movement capability is consisted with change in glass transition temperature (T_g) as seen in Fig. A(a). The VEC concentration dependent τ , B and T_g are all listed in Table B. To investigate the origin of suppressed segment movement with excessive VEC, XRD spectra of SPE with different VEC concentration was measured as seen in Fig. A(b). A broad diffraction peak around 30° was visible when the molar ratio of [IL]:[OFHDODA] to [VEC] was over 8.5:2:5, which was explained by VEC crystallization as proposed by Nano Energy, 2020, 73, 104786.

Table B Summary of τ , B and T_g at different VEC ratio

IL:OFHDODA:VEC	$\tau / \mu\text{s}$	B / K	$T_g / ^\circ\text{C}$
8.5:2:0	8.2	1014	-59
8.5:2:1	6.4	934	-60
8.5:2:3	1.7	825	-63

8.5:2:5	5.0	1028	-56
8.5:2:7	7.6	1040	-54

Overall, the improved ionic conductivity at low VEC concentration is attributed to effective Li⁺ dissociation as well as following facilitated segment movement promoted Li⁺ movement. Nevertheless, VEC crystallization in excessive amount suppressed segment movement, resulting in resisting Li⁺ movement and a lower ionic conductivity as consistent with a higher T_g.

In response to reviewer's critical comment, Fig.A(a) and Table B has been added as Supplementary Fig. 3 and Table 1, Fig.A(b) has been added as Supplementary Fig. 5. The effects of VEC concentration on ionic conductivity have been discussed in Page 8 (in red font color) as follows:

Page 8: "The dielectric spectra of P(IL-OFHDODA-VEC) and P(IL-OFHDODA) was also analyzed as described in Supplementary Note 1. P(IL-OFHDODA-VEC) presented a shorter segment relaxation time (τ) and lower apparent activation energy (B) than that of P(IL-OFHDODA) (Supplementary Table 1). Combined with both DSC and dielectric spectra, it demonstrated..."

Page 8: "However, crystallization of VEC occurred in excessive amount, which was evidenced by a broad diffraction peak around 30° in XRD spectra (Supplementary Fig. 5), resulting in suppressed segment movement and reduced ionic conductivity as consistent with a higher T_g.²⁰"

Fig. A DSC (a), and XRD (b) of SPEs with different VEC ratio.

3) In Fig. 3a, it seems that the current value started to increase around 4.25 V, but the authors set the electrochemical stability window of 5.08 V. Please explain the criteria to set the stability window.

Response: We thank reviewer for the comment. The ESW is read as the voltage of LSV curves where the current exceeds 1.5 μA , which is lower than most literatures (Table C). Furthermore, the LSV is measured with porous carbon electrodes rather than the widely used flat blocking electrodes to avoid overestimate ESW, which has a high contact area between SPE and electrode that is close to composite cathodes.

In response to reviewer's critical comment, the criteria to set the electrochemical stability window has been described in Experimental Section (in red font color) as follows:

Page 10: "Fig. 3a shows the LSV of an asymmetrical cell of $\text{Li|P(IL-OFHDODA-VEC)|carbon}$ at 25 $^{\circ}\text{C}$, wherein ESW $\sim 5.08\text{ V}$ (vs. Li^+/Li) is read as the voltage at which the oxidation current exceeds 1.5 μA . Here, the porous carbon electrode rather than the flat blocking electrode was used to avoid overestimating the ESW, which has a high contact area that is close to composite cathodes."

Table C Comparison of the criteria to set the ESW

Solid Polymer Electrolyte	Response current of ESW / μA	Ref.
P(IL-OFHDODA-VEC)	1.5	This work
PEGDME-4	5	Energy Environ. Sci., 2020, 13, 1318-1325
LATP/PVDF-TiFE/ILE	15	Adv. Energy Mater., 2021, 2101339

4) Please calculate the charge passed through from the Galvanostatic cycling in Fig. 4a, which would help to compare the result with the literature value.

Response: We thank reviewer for the comment. The current density 0.1 mA cm^{-2} in Fig. 4a has been converted to charge density 0.1 mAh cm^{-2} based on review's suggestion, which is a widely adopted charge density of the state-of-the-art SPE based LMBs.

In response to reviewer's critical comment, Fig. 4a has been revised.

Page 12: “Fig. 4a shows periodically charged and discharged curves at a charge density of 0.1 mAh cm⁻² in Li||Li symmetric cells.”

5) The shear modulus is one of the important mechanical properties to maintain the interface over the course of plating and stripping. Please characterize the shear modulus of the electrolyte in addition to Young's modulus.

Response: We agree with reviewer’s critical comment that shear modulus is one of the most important mechanical properties that determine anode compatibility. Fig. B shows that the shear modulus of SPE with OFHDODA ~ 0.71 MPa is almost an order of magnitude higher than that without OFHDODA ~ 0.08 MPa.

In response to reviewer’s critical comment, Fig. B has been added as Supplementary Fig. 14. The description of shear modulus of SPE has been added in Page 13 in the manuscript (in red font color) as follows:

Page 13: “Compared with P(IL-VEC), P(IL-OFHDODA-VEC) exhibited not only a higher bulk Young's modulus, but also a higher bulk shear modulus as shown in Supplementary Fig. 14. These results demonstrated that the mechanical strength of SPEs improved significantly by introduction of the crosslinking agent, which was important to maintain the interface over the course of lithium plating and stripping.^{32,33}”

Fig. B Shear modulus of P(IL-OFHDODA-VEC) and P(IL-VEC).

6) It seems the specific capacity of NCM523 batteries significantly decreased at 2C compared to that of LFP. More investigation is needed to prove the feasible application of the electrolyte to the high-potential cathode.

Response: We admire reviewer's expertise in electrochemical performance of lithium batteries. In addition to improving ionic conductivity, electrochemical stability and mechanical strength of the SPE, we found that it is also important to make effects to adjust the loading and compaction of the composite cathode to realize a superior rate performance.

Fig. C shows the discharge capacity of the full cells as the current densities increased from 0.1 C to 2 C before and after NCM523 cathode optimization. The specific capacity at 2 C improved significantly from ~ 84.39 to ~ 101.32 mAh g^{-1} .

In response to reviewer's critical comment, Fig. 6(c) has been replaced by Fig. C.

Fig. C Rate performance of Li|P(IL-OFHDODA-VEC)|NCM523 full cell before and after NCM523 cathode optimization.

Reviewer #2 (Remarks to the Author):

This a **well-written** manuscript presenting analysis of **novel** solid polymer electrolyte formulation for high voltage Li-ion battery. Authors emphasize the use of polyfluorinated cross linkers that improve electrochemical stability and cycleability of electrolyte. The manuscript indeed presents extensive characterization of formulated material and its performance in the battery.

However, conceptually I do not see this approach any different to other approaches where fluorinated polymers were mixed in gel or solid polymer electrolytes. Many of the trends presented are expected, while some of the data, e.g. DFT calculations are just trivial and almost irrelevant. The binding energies of Li with segments in vacuum have little relevance to Li ion interaction in bulk environment. Of course configurations with two O atoms interacting with Li will be energetically more favorable than those with one. If authors wanted to get molecular scale understanding of Li transport mechanism they should consider atomistic molecular dynamics and conduct more systematic analysis of Li coordination, ion-ion correlations, etc. in representative condensed phase.

Response: We thank reviewer's critical comment and respect reviewer's expertise in DFT calculation.

In this work, we proposed a polyfluorinated crosslinking strategy for the first time. Compared with reported polyfluorinated non-crosslinked SPEs, the polyfluorinated crosslinked SPEs exhibited enhanced oxidation resistance for a wider ESW, which was attributed to more chance transmit the inductive electron-withdrawing effect through crosslinked networks to reduce electron density of adjacent oxygen containing polar groups. Combined with high ionic conductivity and mechanical strength, the polyfluorinated crosslinked SPE based full cells with a Li anode and a NCM523 cathode delivers a capacity retention over ~ 90% after 200 cycles with a cutoff voltage of 4.5 V at 0.5C. To the best of our knowledge, this is one of the best capacity retention in lithium metal batteries based on self-supporting SPEs with a cutoff voltage as high

as 4.5 V at room temperature.

We agree with the reviewer that the simulation model of molecules in vacuum is not fully equivalent with the experimental bulk phase. We also agree that atomistic molecular dynamics or *ab-initio* molecular dynamics could yield more details in Li⁺ transport process as comment by the reviewer. On one hand, to fully reconstruct the bulk phase with all the details, extreme large cells must be introduced to eliminate the long-range order of the system as the SPE is not in crystalline phase, but in an amorphous state. More is the complexity to include Li⁺ for investigation of their interaction in such a large system. Practically, many challenges remain in this approach such as atoms coordination identification in experiment is still difficult with the state-of-the-art experimental characterization techniques for accurate large-scale amorphous model building, and description of a mixture of different types of interaction under a single large system remains as a problem for simulation. On the other hand, though they are not exactly the same, the DFT calculation conducted with smaller models are capable of catching the key mechanism of the interaction between Li⁺ and interested SPE segments, which is more suitable via *ab-initio* calculation due to the nature of electron-related properties. Also, the said interaction is also investigated from other angles such as by Raman and ⁶Li NMR spectra in experiment to complete the story.

In response to reviewer's critical comment, the discussion of the origin of ⁶Li NMR fitting peaks has been revised to remove the quantitative correlation between DFT calculated binding energies and ⁶Li NMR spectra measured three emerging peaks with VEC in Page 9 in the manuscript (in red font color) as follows:

Page 9: "Since the interaction between Li⁺ and electron-withdrawing atoms of the functional groups play an important role in the chemical environment of Li⁺, density functional theory (DFT) calculations of the adsorption energy of Li⁺ on different monomers or dimers were carried out. The results are listed in Supplementary Table 2 and optimized geometry models are presented in Supplementary Fig. 6. In general, the Li⁺ adsorbed around the O atom in the VEC side chain (Li⁺-O_{VEC}) exhibited the weakest

binding affinity.

As the weak binding of Li^+ on the O atom in the SPE segment generates a small chemical shift of Li away from 0 ppm, the red fitting peak of ^6Li NMR of P(IL-OFHDODA-VEC) possessed the largest proportion is believed to be the result of the interaction between Li^+ and the O atom in the VEC side chain ($\text{Li}^+\text{-O}_{\text{VEC}}$).³⁰

Manuscript provides plenty of data on cell performance, but not much basic characterization that would inform about mechanisms of observed improvement. For example, authors mentioned influence of fluorinated links on segmental mobility, but the only evidence provided is change in glass transition temperature by few degrees. Mechanical properties are only characterized by Young's modulus. Also DFT calculations confirming that presence of fluorinated segments improves oxidative stability and demonstrating the electron-withdrawing effect would be very useful.

Response: We thank reviewer's critical comment. Based on reviewer's suggestion, we further perform experimental characterizations and DFT calculations to corroborate improved electrochemical stability and mechanical strength of the SPE after polyfluorinated crosslinking.

To further confirm influence of the polyfluorinated crosslinker on segmental mobility, the τ and B at different OFHDODA ratio was compared. Both the τ and B increased with OFHDODA ratio (Table D), which are consisted with the higher T_g in Fig. D, indicating suppressed segment movement.

Table D Summary of τ , B and T_g at different OFHDODA ratio

IL:OFHDODA:VEC	$\tau / \mu\text{s}$	B / K	T_g
8.5:0:3	/	/	-66
8.5:2:3	1.7	825	-63
8.5:4:3	10.9	1298	-48
8.5:6:3	75.6	1540	-43

Fig. D DSC of SPEs with different OFHDODA ratio.

In addition to improved Young's modulus, shear modulus, another important mechanical property, has also been improved from 0.08 MPa without crosslinker to 0.71 MPa with crosslinker as shown in Fig. B, which benefits anode interface compatibility over the course of lithium plating and stripping.

To confirm the improved oxidative stability of SPE with polyfluorinated crosslinker, DFT calculations have been performed to investigate the effect from the polyfluorinated segment and the ability of crosslinking separately. The former is studied with polyfluorinated crosslinker OFHDODA and nonfluorinated crosslinker HDODA, while the latter is studied with polyfluorinated crosslinker OFHDODA and polyfluorinated non-crosslinker OFPMA. For comparability, all three molecules are copolymerized with two VEC molecules.

Firstly, the electron-withdrawing effect of fluorinated segments is studied by DFT calculations. The partial charge distribution of OFHDODA, HDODA and OFPMA are calculated. It is found that total charges of fluorene atoms ($\sim -0.570 - -0.592$) in

OFHDODA are more negative than that of hydrogen atoms (0.004 – 0.054) on HDODA as shown in Fig. E. While total charges of fluorene atoms (~ -0.570 – -0.589) in OFPMA are comparable with that in OFHDODA.

Fig. E DFT calculated partial charge distribution of the OFHDODA (a), HDODA (b) and OFPMA (c).

Fig. F HOMO of the OFHDODA (a), HDODA (b) and OFPMA (c) copolymerized with two VEC.

Table E Ionization energy of the OFHDODA, HDODA and OFPMA copolymerized with two VEC

Trimer	Neutral state energy / eV	Cation state energy / eV	Ionization energy / eV
VEC-OFHDODA-VEC	-67605.44	-67596.24	9.19
VEC-HDODA-VEC	-45996.44	-45987.47	8.97
VEC-OFPMA-VEC	-59262.13	-59253.22	8.91

Secondly, HOMO plots of the three trimers are shown in Fig. F. The frontier orbital calculation shows a more delocalized HOMO for OFHDODA case. Due to electron withdrawing effect from the fluorinated segments, the ester segment is less activated and due to the ability of cross-linking, the VEC segment on both sides of the molecule would not attract too much electron to them. As a result, the electron in the frontier orbital is distributed between the ester segment and the VEC segment. For the HDODA case, the lack of strong electron-withdrawing fluorinated segments would leave the ester group with all the frontier orbital, leading to a possible easier oxidization position. For the OFPMA case, the lack of the ability of cross-linking seems lead to frontier electron concentration on the VEC segments which may result in another possible oxidization position.

Thirdly, the value of the HOMO level is as follows: OFHDODA trimer \sim -8.26 eV, HDODA trimer \sim -7.83 eV and OFPMA trimer \sim -7.95 eV. The OFHDODA case has the deepest HOMO level. Also, the energy requirement to remove an electron for these three trimers are 9.19 eV for OFHDODA trimer, 8.97 eV for HDODA trimer, 8.91 eV for OFPMA trimer respectively, indicating that the OFHDODA case is the most stable one (Table E). All these studies indicate that OFHDODA with both polyfluorinated segment and the ability to crosslinking is the most resilient to oxidization, which are consisted to the widest ESW of the SPE with OFHDODA.

In response to reviewer's critical comment, Fig. D has been added as Supplementary Fig. 4, Fig. E has been added as Supplementary Fig. 8, Fig. F has been added as Supplementary Fig. 9 and Table E has been added as Supplementary Table 4. The discussion of effects of polyfluorinated crosslinker on segment relaxation time,

shear modulus and oxidative stability of SPE has been added in Page 11 and 13 in the manuscript (in red font color) as follows:

Page 11: “The effect of OFHDODA on ESW of the SPE can be understood from the molecular structure. OFHDODA possesses both the difluoro methylene chain and the ability of crosslinking. From DFT calculation, the difluoro methylene chain in OFHDODA has a strong electron-withdrawing effect, which effectively reduces the electron density of its neighbor ester groups to enhance the overall oxidation resistance, especially after copolymerized with VEC groups (Supplementary Fig. 8 and 9). After coupled with VEC groups, the electron-withdrawing effect from the polyfluorinated group can suppress the frontier orbital electron from concentrating on the ester group but also distributes on VEC segments. Delocalization of frontier electron is helpful for avoiding easy oxidation on a single reaction position. Such behavior is absent in nonfluorinated crosslinker HDODA site. The HOMO is concentrated solely on ester groups after VEC copolymerization for HDODA. As a second proof, the required energy to remove an electron from the system for VEC copolymerized OFHDODA ~ 9.19 eV is higher than that for VEC copolymerized HDODA ~ 8.97 eV (Supplementary Table 4). Furthermore, the ESW of SPE with HDODA reduced to 4.55 V (vs. Li⁺/Li, Supplementary Fig. 10) is another proof of stabilization effect of polyfluorinated OFHDODA group.

The effect from the ability of crosslinking is also studied. After replacing OFHDODA by fluorinated non-crosslinkable OFPMA, it is observed that the frontier electron is concentrated on one of the VEC segment after copolymerization as seen in Supplementary Fig. 9, suggesting less stability than OFHDODA case. The energy requirement of removing an electron from the system agrees with a value of 8.91 eV for VEC copolymerized OFPMA (Supplementary Table 4). The ESW of SPE with OFPMA ~ 4.71 V (vs. Li⁺/Li) was slightly higher than that with HDODA but lower than that with OFHDODA (Supplementary Fig. 10). The non-crosslinking nature might work in the opposition of stabilization, as all the copolymerization has to take place on one side of the molecule, resulting VEC segments easier be attacked than the ester

group of OFPMA that being surrounded by VEC segments. In addition, the crosslinked structure is more bulky and less linear shaped. This may inhibit sidechain movement and reduce possibility of contact between oxygen containing polar groups and electrodes induced oxidation as supported by a higher T_g in DSC, higher B and longer τ in dielectric spectra (Supplementary Fig. 4 and Table 5).”

Page 13: “Compared with P(IL-VEC), P(IL-OFHDODA-VEC) exhibited not only a higher bulk Young's modulus, but also a higher bulk shear modulus as shown in Supplementary Fig. 14. These results demonstrated that the mechanical strength of the SPE improved significantly by introduction of the crosslinking agent, which was important to maintain the interface over the course of lithium plating and stripping.^{32,33}”

Reviewer #3 (Remarks to the Author):

In this study, authors developed a new solid polymer electrolyte composed of a fluorinated cross-linker, an ionic liquid-type monomer, and vinyl ethylene carbonate (VEC). As noted in the introduction, the high conductivity of PVECs (ex. Nano Energy, 2020, 73, 104786), and the improvement of oxidation resistance of SPEs by using fluorine monomers (ex. Adv. Funct. Mater. 2021, 31, 2101736), have already been reported. Therefore, in my opinion, the novelty of the paper is not enough high, as the concept of this study is kind of combination of previous studies. However, the high oxidation resistance as well as high ionic conductivity of this SPE achieved **excellent battery performance** comparable to those of state-of-the-art SPEs, as shown in Table S2. For this reason, the paper might be **interest in the readership of Nature Communications**. However, major and minor points need to be addressed prior to publication.

1. Long-term cycling tests for symmetric Li cells were performed only at a very low current density of 0.1 mA cm^{-2} despite the high ionic conductivity of the SPE. For practical application of the SPE toward Li metal batteries, cycling performance at higher current densities such as 1 mA cm^{-2} should be investigated.

Response: We thank reviewer for the comment. Based on reviewer's suggestion, we have performed cycling tests for symmetric Li cells at a current density of 1.0 mA cm^{-2} as seen in Fig. G, which exhibits a flat polarization curve with a constant polarization as low as 390 mV for up to 100 h.

Fig. G Galvanostatic cycling of lithium symmetrical cells with P(IL-OFHDODA-VEC) at 1.0 mA cm⁻².

2. I don't understand why there is the optimal ratio for ionic conductivity among [IL], [VEC], and [OFHDODA]. In particular, the fluorine cross-linker would not contribute to ionic conductivity. Author should address the mechanism of the optimized monomer ratio. In addition, the role of the IL monomer should be clarified. The fluorine cross-linker contributes to the oxidative stability and the VEC to conductivity, then what is the role of the IL monomer?

Response: We appreciate reviewer's critical comment. To realize a high ionic conductivity, the SPE will not only effectively dissolve lithium salts and dissociate Li⁺ from TFSI⁻, but also efficiently conduct Li⁺ via polar group coordination and chain segment movement. The IL, VEC, and OFHDODA will contribute to these process as follows:

- (1) The high polarity IL offers better solubility of lithium salts than that of PEO as proposed by Joule 2019, 3, 2687-2702.
- (2) The interaction between VEC and Li⁺ effectively dissociates Li⁺ from TFSI⁻ as proved by Raman spectra and solid-state NMR in Fig. 2.
- (3) The ether groups in VEC and OFHDODA show high affinity with Li⁺ as supported by DFT calculations in Supplementary Fig. 6 and solid-state NMR spectra in Fig. 2. Thus the ester groups in VEC and OFHDODA are believed to be the hopping site for Li⁺, as the Li⁺ coupling/decoupling process is likely to take place around them.

(4) The segment movement capability can be maintained if over-crosslinking and VEC crystallization can be prevented, which is evidenced by DSC in Fig. A(a) and D and XRD in Fig. A(b) .

Therefore, the ratio of IL, VEC, and OFHDODA monomers needs to be deliberately adjusted to realize an optimized condition.

In addition to dissolve lithium salts, the ILs with superior flaming resistance benefits high safety of SPE based LMBs. Furthermore, the pyrrole IL used here does not contain unsaturated bonds, which exhibits better oxidation resistance than that of widely used imidazole ion liquid, resulting in a wide electrochemical stability window to improve high-voltage stability.

In response to reviewer's critical comment, the effects of monomer ratio on ionic conductivity of SPE and the role of IL have been discussed in Page 5 and 9.

Page 5: "The SPE with **flaming resistant** IL is nonflammable as shown in Supplementary Video 1."

Page 9: "**Overall, to realize a superior ionic conductivity, the SPE will not only effectively dissolve LiTFSI and then dissociate Li⁺ from TFSI⁻, but also efficiently coordinate Li⁺ and promote its mobility. The high polarity IL offers good solubility of LiTFSI.³¹ The interaction between the O atom in the VEC and Li⁺ effectively dissociates Li⁺ from TFSI⁻ as proved by the Raman spectra. The ester groups show high affinity with Li⁺ as supported by DFT calculations and solid-state NMR spectra. Thus the ester groups are believed to be the hopping site for Li⁺, as the Li⁺ coupling/decoupling process is likely to take place around them. The segment movement capability can be maintained to prevent over-crosslinking and VEC crystallization, which is evidenced by XRD, DSC and dielectric spectra. Therefore, the ratio of IL, VEC, and OFHDODA monomers needs to be deliberately adjusted to achieve an optimized condition.**"

3. It is surprising that the SPE shows such a high conductivity over 1 mS/cm. I am afraid that there are unreacted monomers that act as plasticizer of the SPE. Since the FT-IR-ATR could observe only a few micrometers of the surface of the sample, it might be overstatement that all monomers are fully reacted. I think it is necessary to clarify the complete reaction of all monomers by other methods such as an extraction test of unreacted monomers using organic solvents.

Response: We appreciate reviewer's comment on the monomer polymerization degree. Based on reviewer's suggestion, we have performed an extraction test of SPE using diethyl ether, which can dissolve unreacted monomers and oligomers. It exhibited a small loss in weight ~3.1 %, which was mainly attributed to dissolved oligomers with molecular weight range from 751 to 1255 rather than unreacted monomers as seen in Fig. H.

In response to reviewer's comment, Fig. H has been added as Supplementary Fig. 2. The result of extraction test of SPE using diethyl ether has been described in Page 6 (in red font color).

Page 6: "Combined with both FTIR spectra and TGA, it demonstrated that IL, OFHDODA and VEC monomers of the SPE were copolymerized by the C=C bonds after UV light curing, which was confirmed by a minor weight loss of ~3.1 % due to oligomers extracted by diethyl ether (Supplementary Fig. 2)."

Fig. H Mass spectrum of oligomers of the SPE extracted by diethyl ether.

4. For readers' information, the thickness of the SPE should be described.

Response: We thank reviewer for the comment. The thickness of the SPE is $\sim 100 \mu\text{m}$.

In response to reviewer's comment, the thickness of the SPE has been described in Page 5.

Page 5: "The SPE film $\sim 100 \mu\text{m}$ in thickness consisted of three monomers..."

5. Authors conducted the AFM measurement to investigate the elastic modulus of the SPE. However, the information obtained by AFM is the elastic modulus of the sample surface. Therefore, it would be better to investigate the bulk mechanical properties by tensile testing or DMA measurements.

Response: We agree reviewer's critical comment that bulk mechanical properties of the SPE are important for anode interface compatibility. We have investigated bulk mechanical properties of the SPE based on reviewer's suggestion.

Fig. I shows bulk Young's modulus of the SPE with OFHDODA, which is ~ 115 MPa averaging three sites. While the bulk Young's modulus of the SPE without OFHDODA is lower than detection limit. Fig. B shows bulk shear modulus of the SPE,

which increases from 0.08 MPa to 0.71 MPa after incorporation of OFHDODA. All these results confirm improved bulk mechanical properties of SPE after crosslinking.

In response to reviewer's critical comment, Fig. I has been added as Supplementary Fig. 13. The description of improved bulk mechanical properties of the SPE has been added in Page 13 in the manuscript (in red font color) as follows:

Page 13: "Since the information obtained by AFM came from the sample surface, the bulk mechanical properties were further measured (Supplementary Fig. 13). Compared with P(IL-VEC), P(IL-OFHDODA-VEC) exhibited not only a higher bulk Young's modulus, but also a higher bulk shear modulus as shown in Supplementary Fig. 14. These results demonstrated that the mechanical strength of SPEs improved significantly by introduction of the crosslinking agent, which was important to maintain the interface over the course of lithium plating and stripping.^{32,33}"

Fig. I Bulk Young's modulus of three different sites of SPEs with OFHDODA.

REVIEWERS' COMMENTS

Reviewer #1 (Remarks to the Author):

The authors have satisfactorily addressed all the comments from the reviewer and no further revision is needed.

Reviewer #3 (Remarks to the Author):

I am satisfied with the revision.

REVIEWER COMMENTS

Reviewer #1 (Remarks to the Author):

The authors have satisfactorily addressed all the comments from the reviewer and no further revision is needed.

Response: We appreciate reviewer for support and help.

Reviewer #3 (Remarks to the Author):

I am satisfied with the revision.

Response: We thank reviewer for support and help.